# Role of Oxidative Stress and Carcinoembryonic Antigen-Related Cell Adhesion Molecule 1 in Nonalcoholic Fatty Liver Disease

**DOI:** 10.3390/ijms241411271

**Published:** 2023-07-10

**Authors:** Plator Memaj, Zayd Ouzerara, François R. Jornayvaz

**Affiliations:** 1Division of Endocrinology, Diabetes, Nutrition and Therapeutic Patient Education, Department of Medicine, Geneva University Hospitals, 1205 Geneva, Switzerland; plator.memaj@hcuge.ch (P.M.); zayd.ouzerara@hcuge.ch (Z.O.); 2Diabetes Center, Faculty of Medicine, Geneva University, 1205 Geneva, Switzerland; 3Department of Cell Physiology and Metabolism, Faculty of Medicine, Geneva University, 1205 Geneva, Switzerland

**Keywords:** NAFLD, NASH, insulin resistance, diabetes, oxidative stress, CEACAM1

## Abstract

Nonalcoholic fatty liver disease (NAFLD) has become a widely studied subject due to its increasing prevalence and links to diseases such as type 2 diabetes and obesity. It has severe complications, including nonalcoholic steatohepatitis, cirrhosis, hepatocellular carcinoma, and portal hypertension that can lead to liver transplantation in some cases. To better prevent and treat this pathology, it is important to understand its underlying physiology. Here, we identify two main factors that play a crucial role in the pathophysiology of NAFLD: oxidative stress and the key role of carcinoembryonic antigen-related cell adhesion molecule 1 (CEACAM1). We discuss the pathophysiology linking these factors to NAFLD pathophysiology.

## 1. Introduction

Nonalcoholic fatty liver disease (NAFLD) is a serious disease that affects 25% of the world’s population [1]. The epidemic of overweight and obesity has also contributed to an increase in the prevalence of NAFLD over the past couple of years [2]. This is an important pathology to understand because it can lead to more concerning diseases like advanced hepatic fibrosis, nonalcoholic steatohepatitis (NASH), or even hepatocellular carcinoma [3]. Due to the prevalence and associated complications of NAFLD, it is essential to analyze the diverse physiological mechanisms for targeted and effective treatment.

NAFLD is mostly encountered in individuals with metabolic syndrome, with both type 2 diabetes and obesity being the main risk factors [1]. Those diseases come mostly from bad habits in food consumption, especially hypercaloric food, and from lack of physical activity.

The gold standard for NAFLD diagnosis remains the histological evaluation of a liver biopsy [4]. Imaging, such as magnetic resonance imaging and abdominal ultrasound, play a crucial role to guide the physician to suspect and diagnose NAFLD.

The underlying mechanisms of NAFLD development are complex, and multifactorial constellation of biochemical reactions are linked to increased intrahepatic fat accumulation and inflammation [5].

The main therapeutical approach relies on lifestyle management, but since our understanding of NAFLD is increasing, new potential molecules seem to have promising results [6,7]. The molecular comprehension of the processes leading to NAFLD remains decisive to understand and to find new therapeutical targets.

This study will focus on two significant processes that contribute to the pathogenesis of NAFLD, namely oxidative stress and carcinoembryonic antigen-related cell adhesion molecule 1 (CEACAM1). CEACAM1 is a transmembrane protein that plays a crucial role in insulin degradation and excretion, as well as regulating insulin homeostasis in the liver. The redox equilibrium must remain within physiological parameters in order to be considered safe. However, if exposed to certain triggers, a redox imbalance may arise, leading to an increase of oxidative stress and ultimately contributing to the pathogenesis of NAFLD.

## 2. Method

Several articles were included in a literature review that was conducted using PubMed, Google Scholar, and Web of Science from 1986 until 2023, but mostly from recent years, which were associated with the link between oxidative stress, CEACAM1, and MAFLD or NAFLD. Medical Subject Headings terms such as “Nonalcoholic fatty liver disease”, “Liver disease”, “NASH”, “Steatohepatitis”, “Steatosis”, “Hepatic Lipogenesis” were associated with “CEACAM1”, “Insulin clearance”, and “Oxidative Stress”. The different publications were evaluated and chosen based on their abstract significance. The research sites’ suggestions for similar articles were additionally considered and chosen. In total, this review was based on the study of 94 different articles. The analyzed articles were all restricted to English language.

### 2.1. NAFLD: Definition and Prevalence, Diagnosis, Pathophysiology, and Therapeutical Approach

#### 2.1.1. Definition and Prevalence

NAFLD is a spectrum of diseases ranging from simple hepatic steatosis to nonalcoholic steatohepatitis and hepatic fibrosis. It is a very common disease and the most common hepatic disease worldwide with an estimated prevalence of 25% [1]. It is defined by hepatic steatosis (fat in at least 5% of hepatocytes) and the absence of a secondary cause of hepatopathy such as excessive alcohol consumption, chronic infection with viral hepatitis, and other chronic liver diseases [8,9,10]. Liver biopsy is the gold standard for NAFLD diagnosis, as well as for NASH and hepatic fibrosis [11]. Histologically, NASH, which is a subcategory of NAFLD, is characterized by the presence of hepatic steatosis, liver inflammation, and hepatocellular ballooning on biopsy in the absence of any chronic liver disease or alcohol consumption [11,12]. NASH prevalence in the general population is estimated to range from 1.5% to 6.4% [1].

#### 2.1.2. Risk Factors

NAFLD prevalence increases dramatically in type 2 diabetic and obese patients [13]. It has been estimated that approximately 70% of individuals suffering from type 2 diabetes mellitus (T2D) have NAFLD, with an additional 25% presenting with a more severe manifestation known as nonalcoholic steatohepatitis (NASH) [14,15]. Although evidence is limited, these patients appear to be at a higher risk of developing advanced fibrosis and hepatocellular carcinoma [16]. The prevalence of NAFLD in individuals with obesity has been estimated to range from 60–95% [17,18,19]. The presence of sleep apnea syndrome is independently associated with NAFLD, degree of steatosis, inflammation, and fibrosis [20]. NAFLD is also associated with consumption of western foods, including soft drinks, red meat, processed foods, and refined grains [21,22].

#### 2.1.3. Diagnosis of NAFLD

The diagnosis of NAFLD can be confirmed through histological examinations post-biopsy or through imaging modalities such as magnetic resonance imaging (MRI) or ultrasound. Liver biopsy remains the gold standard for identifying and categorizing histological changes in NAFLD [4,23]. As mentioned above, NAFLD is defined histologically by at least 5% hepatic fat content. Hepatocellular damage, lobular inflammation, and hepatocellular ballooning are three characteristics of NASH in the context of NAFLD [24].

Given the potential risks associated with liver biopsy, imaging is generally the favored method for detecting NAFLD. Computed tomography (CT), MRI, and ultrasonography are common techniques employed to quantify hepatic fat in the body; however, these conventional imaging modalities can be limited due to a lack of sensitivity and specificity (in CT and ultrasonography), subjectivity (in MRI and ultrasonography), potential radiation-related hazards (in CT), and other confounding factors [25]. One of the main sources of confusion is differential diagnosis, especially hepatic glycogenesis and glycogenic hepatopathy [26]. However, current developments in imaging, such as multi-parametric MRI, can aid in more effectively detecting hepatic fat. Indeed, multi-parametric MRI, particularly with the proton density fat fraction, has been developed. As a result, it has evolved into a virtual liver biopsy method that may be used to monitor patients during treatment and can prevent needless biopsies [25]. This new imaging technique may be essential given the high and rising frequency of NAFLD.

#### 2.1.4. Pathophysiology

The progression of NAFLD is caused by several underlying factors, including accumulation of intrahepatic fat, adipose tissue malfunction, and de novo intrahepatic lipogenesis. Ultimately, this process is mainly attributed to insulin resistance [27]. Additionally, some lipid intermediates involved in NAFLD development, such as diacylglycerols and ceramides, have a higher propensity than others to lead to hepatic insulin resistance, feeding a vicious loop that results in a rise in NAFLD [28]. In fact, increasing levels of free fatty acids (FFA) in the blood and ectopic lipid buildup in the liver are related to insulin resistance, which can further encourage inflammation and endoplasmic reticulum stress, contributing to the vicious cycle of the insulin resistant state [29]. Inflammation appears to play a substantial role in both insulin resistance and NAFLD because inflammatory mediators like cytokines and adipokines play a vital function not only in inflammation but also in metabolic energy balance and immunological response [5]. An important contributing element to the development of NAFLD is the increase of intracellular reactive oxygen species (ROS), which produce oxidative stress. The primary ROS generators are NADPH oxidase (NOX) enzymes, and it has been demonstrated that NAFLD and insulin resistance are associated with increased NOX activity because of hepatic lipid overload [30,31]. Additionally, it is well-known that obesity and poor eating practices increase the formation of ROS by disrupting the balance between their production and clearance, which further contributes to the emergence of insulin resistance and liver tissue damage, perpetuating the vicious cycle [30]. Also, subjects with lower plasma adiponectin concentrations are more likely to develop NAFLD than those with normal adiponectin levels since adiponectin is known to have anti-inflammatory characteristics and promote hepatic insulin sensitivity [32].

#### 2.1.5. Therapeutical Approach

Lifestyle management is the foundation of NAFLD treatment, and mostly consists of improving diet and increasing exercise. Studies have shown that weight reduction of 7–10% is associated with a decrease in NAFLD-related inflammation [6]. A study has shown that bariatric surgery in the setting of weight loss results in histologic improvement of steatosis, steatohepatitis, and liver fibrosis in most patients with NAFLD [33]. Due to the importance of insulin resistance in the pathophysiology of NAFLD, medications that increase insulin sensitivity have been studied. Metformin has not shown convincing results in patients with NAFLD [34]. Statins are commonly taken by patients with NAFLD, but there are no studies demonstrating their efficacy in NAFLD. As for antioxidant therapies, vitamin E therapy tends to have good results, but its long-term efficacy is still not well known [7]. Glucagon-like peptide-1 (GLP-1) receptor agonists are a type of medication that can cause dose-dependent weight loss. Additionally, these drugs have been demonstrated to improve insulin resistance in the liver and adipose tissue, reduce de novo lipogenesis and lipolysis in fat tissue, and reduce oxidative stress [35,36,37,38,39]. In a randomized, double-blind, placebo-controlled, multicenter study lasting 48 weeks, liraglutide (a GLP-1 receptor agonist) was found to treat nonalcoholic steatohepatitis (NASH) effectively and safely, leading to histological improvement [38]. One study has shown that semaglutide, another GLP-1 receptor agonist, improves liver steatosis in patients with T2D [40]. In another study conducted in mouse models, semaglutide was shown to reduce fat accumulation in the liver and to play an anti-inflammatory role in advanced NAFLD in mice with NAFLD [41].

Liver transplantation is still the only therapeutic option for people with end-stage NAFLD-related liver disease. [42]. In fact, fibrosis is a determining factor for predicting survival and liver transplantation [43].

### 2.2. Oxidative Stress

#### 2.2.1. Generalities

Oxidative stress refers to several harmful processes that result from an imbalance between excessive formation of prooxidants and inadequate antioxidant defenses [44]. This imbalance leads to cell death and tissue damage. Classic prooxidants are ROS (superoxide, hydrogen peroxide, hydroxyl radical) and reactive nitrogen species (oxide nitrate, peroxynitrite). Under physiological conditions, ROS continue to be formed but are eliminated by cell scavenging systems (catalase, superoxide dismutase, glutathione). Studies have identified several circulating biomarkers of lipid peroxidation in patients with NAFLD/NASH [45,46,47,48] and disease severity was found to be at high levels of these markers [49]. More than 90% of patients with NAFLD were found to have an imbalance of oxygen reduction and decreased plasma antioxidant capacity [50].

ROS are generated at various sites, such as the mitochondrial respiratory chain, cytochrome P450, oxidative enzymes, and some heme proteins. Oxidation of fatty acids in the liver is a reaction that is one of the main sources of ROS. Beta-oxidation occurs in the mitochondria as well as in the peroxisome, whereas omega-oxidation occurs in the endoplasmic reticulum by cytochrome p450 [51].

#### 2.2.2. Oxidative Stress in NAFLD Development

Although poorly understood, the progression of hepatic steatosis to NASH was initially explained by the two-hits hypothesis. The first cause is hepatic steatosis resulting from insulin resistance, which leads to lipogenesis in the liver and less than optimal export of fatty acids [52,53] and hormonal disorders such as the alteration of adiponectin and leptin [54,55]. When insulin resistance first occurs in muscle and adipose tissue, less glucose is delivered to these tissues, creating a catabolic state that results in the release of fatty acids into the circulation through peripheral fat. To compensate for hyperglycemia, the pancreas secretes more insulin and hyperinsulinemia results. Since the liver is relatively more sensitive to insulin, it enters an anabolic state and secretes and stores lipids [56]. The second hit is triggered by two types of factors: those that increase inflammatory cytokines that activate stellate cells and cause fibrosis and endoplasmic reticulum stress, and those that increase oxidative stress [57,58,59,60,61]. Endoplasmic reticulum stress is involved in the development of steatosis and NASH [62]. In fact, ceramides, but also sphingolipids derived from saturated fatty acids, have been shown to be synthesized when saturated fatty acid content is increased. Ceramides induce hepatocyte apoptosis through endoplasmic reticulum stress and may induce mitochondrial fragmentation through interaction with mitochondrial fragmentation factor [63]. In addition, a large stream of free fatty acids, folded and unfolded proteins accumulate, leading to a process called unfolded protein response. The latter leads to a stress of the endoplasmic reticulum and thus to the production of ROS [64].

ROS can attack molecules such as polyunsaturated fatty acids in the cell, producing aldehyde by-products such as 4-hydroxy-2-ninenal and malondialdehyde [65,66]. Since the mitochondrial membrane is composed of polyunsaturated fatty acids for the proper assembly of respiratory complexes, the production of these two aldehyde products impairs mitochondrial function. A study has shown that ROS can directly attack hepatic mitochondrial DNA, leading to a decrease in the synthesis of proteins important for the respiratory chain. It is worth noting that a new theory of “multiple hits” has more recently emerged (Figure 1). It hypothesizes that, in addition to the two steps described above, there are genetic and epigenetic mutations and factors from the gut microbiota that would play a role in the pathogenic development of NAFLD [67]. Finally, one study found that production of ROS, lipid peroxidation, alteration of the mitochondrial respiratory chain, and changes in mitochondrial membrane composition are present in all patients with hepatic steatosis and tend to progress to NAFLD [68].

At the same time, nutrition also plays an important role. Chronic malnutrition, especially based on high fat and low glycemic index carbohydrates, can stimulate intracellular liver pathways leading to the development of oxidative stress. One study has shown that saturated fat consumption plays an important role in the development of steatosis and NAFLD. Another study found that the antioxidant components of the diet were significantly reduced in patients with NASH compared with healthy people [69]. There are mechanisms to balance the accumulation of triglycerides and fatty acids in patients with a high-fat diet. Thus, there is an increase in mitochondrial palmitoyl transfer carnitine and mitochondrial uncoupling proteins, which lowers intracellular levels of ROS and thereby plays a protective role in the development of NAFLD [63].

A decrease in the antioxidant activity of catalase and superoxide dismutase has also been observed [70]. Indeed, ROS, produced in NAFLD, can degrade antioxidant molecules and inhibit antioxidant enzymes [71]. In addition, nuclear factor-2, a transcriptional regulator of antioxidant proteins, is completely degraded, further worsening the redox balance [72].

### 2.3. Carcinoembryonic Antigen-Related Cell Adhesion Molecule 1 (CEACAM1)

#### 2.3.1. Definition and Generalities

Carcinoembryonic antigen-related cell adhesion molecule 1 (CEACAM1) is a transmembrane glycoprotein, a member of the immunoglobulin superfamily and, more precisely, it belongs to the carcinoembryonic antigen (CEA) family [73]. In the liver, CEACAM1 is essentially expressed on epithelial cells and endothelial cells participating in the conservation of the epithelial polarity in hepatocytes during their differentiation. Originally, in humans the gene encoding CEACAM1 protein is located on the long arm of the chromosome 19 [74].

When it comes to splicing, human CEACAM1 gene generates 12 splice variants [75]. Given this diversity in splicing, it is not a surprise that it has a lot of different roles which can differ according to its location and its stimulus. Its functions include controlling immunological response, metastasis, tumor suppression, angiogenesis, apoptosis, and the layout of tissue structure [73]. It is important to note that the major CEACAM1 isoforms are made of 1 transmembrane domain with 3 or 4 extracellular domains, including the N-terminal IgV-like domain, and a cytoplasmic tail, which can be short or long. To resume, there are CEACAM1-4L and the CEACAM1-4S, which both have 4 extracellular domains but with a long cytoplasmic tail and a short one, respectively, and CEACAM1-3L and the CEACAM1-3S, which have 3 extracellular domains with a long cytoplasmic tail and a short one, respectively [73].

#### 2.3.2. CEACAM1 Role in Metabolic Balance and NAFLD/NASH

Here, we focus on the molecular aspects of the role of CEACAM1 in the development of nonalcoholic fatty liver disease, including NAFLD, NASH, and liver fibrosis.

First, this protein is known to have a key role in hepatocyte differentiation, especially in early life where CEACAM1 is more expressed in hepatocytes. A major role in hepatic regeneration has also been noticed in rat livers after partial hepatectomy where CEACAM1 expression was analyzed through indirect fluorescence [76]. On the genetic aspect, CEACAM1 expression is essentially upregulated by the hepatocyte nuclear factor 4α (HNF4α) and CEACAM1’s promoter studies in rats have shown to have binding sites for factors including cAMP-response element binding protein (CREB), CCAAT/enhancer binding protein (C/EBP), hepatonuclear factor 5 (HNF5), glucocorticoids, hepatonuclear factor 1 (HNF1), and activator protein 1 and 2 (AP-1 and AP-2) [77]. CEACAM1 acts as a tumor suppressor, in tissue organization and in hepatocyte differentiation. These different functions have been known for a long time but CEACAM1’s role in metabolic processes, although suspected, has only recently been studied and explored.

Indeed, CEACAM1 plays a central function in insulin transduction since it has been reported as a substrate of the insulin receptor specifically in the liver, in opposition to the other insulin-sensitive tissues (adipose tissue and skeletal muscle) [78].

Once insulin reaches the hepatocyte, it enhances CEACAM1 action on the hepatocyte mitogenic activity by downregulating it. In fact, the action of CEACAM1, phosphorylated by the insulin receptor, is allowed through its binding to SH2-containg adapter protein (Shc), which becomes sequestered by CEACAM1. Since Shc is a coupler protein, this has the effect of reducing the coupling of the growth factor receptor-bound protein 2 (Grb2) to the insulin receptor and inhibits the mitogenesis pathway mediated by Ras/MAP kinase resulting in a reduced mitogenic activity. Another consequence of the Shc sequestering by CEACAM1 is the reduced activity of the phosphoinositide 3′kinase (PI3K) and protein kinase B (AKT). This pathway is usually made possible through binding between insulin receptor substrate 1 (IRS-1) and PI3k after IRS-1 is phosphorylated by the insulin receptor and leads to cell proliferation. CEACAM1’s binding to Shc competes with IRS-1 phosphorylation by the insulin receptor, hence downregulating cell proliferation in hepatocytes [79].

Regarding insulin clearance, CEACAM1 plays a central role, orchestrating multiple intracellular reactions initiated by insulin binding to its receptor. This binding causes CEACAM1 phosphorylation, allowing its indirect attachment to the insulin receptor in order to enhance endocytosis, degradation, and clearance of the insulin–insulin receptor complex [80].

SHP-2 is a tyrosine phosphatase which is a protein that dephosphorylates and inhibits insulin receptor substrate 1 (IRS-1) action. CEACAM1 attaches itself to the SHP-2 tyrosine phosphatase, preventing its negative action on IRS-1, which results in a sustained action of IRS-1 in the hepatocyte. IRS-1 sustained action is also favored by the insulin–insulin receptor complex endocytosis because of insulin receptor tyrosine kinase prolonged exposure to IRS-1 [80,81].

These mechanisms point out the major role of CEACAM1 in insulin degradation and in insulin clearance in hepatocytes since hepatic insulin removal is responsible for clearing approximatively 80% of the total insulin synthetized by pancreatic β cells [82]. Mutation on the phosphorylation site in CEACAM1 has been shown to cause hyperinsulinemia, glucose resistance, and leads ultimately to chronic fatty hepatopathy in mouse models [83].

The role and impact of CEACAM1 protein in the development of chronic NAFLD is intimately linked to its function as an insulin clearer. In non-mutated individuals, it has been hypothesized that chronic hyperinsulinemia and impaired insulin secretion pulsatility reduce CEACAM1 phosphorylation and action, creating a vicious cycle [84,85]. There is strong evidence that NAFLD patients’ hyperinsulinemia is significantly more correlated with poor insulin clearance than with elevated insulin production [86], implying that CEACAM1 is one of the major key factors linking insulin resistance, hyperinsulinemia, and fatty liver disease.

Insulin is known to increase the production and uptake of cholesterol, FFA, triglycerides, and phospholipids through enhancing sterol regulatory element-binding proteins (SREBPs) in hepatocytes [87]. SREBPs are also essential for enzymes expression, for example, glucokinase, liver-type pyruvate kinase (LPK), fatty acid synthase (FAS), and acetyl-CoA-carboxylase (ACC), which are required for lipogenesis [88]. Insulin concentration is much higher in the portal vein than in the peripheral blood circulation [82], therefore the liver seems to be more likely to increase de novo lipogenesis through insulin action. However, despite the liver expressing more lipogenic genes, the activity of fatty acid synthase (FAS) in normal insulinemic individuals is nearly imperceptible and it is mainly due to insulin release pulsatility [85]. This allows an acute and fast action of insulin on the hepatocytes, phosphorylating CEACAM1 protein, permitting insulin clearance through endocytosis of the insulin–insulin receptor complex. As a reminder, insulin is usually secreted by β cells in two phases: a first secretion peak phase following glucose intake, then a second deferred phase that helps bring glycemia back to normal blood concentrations [89]. In individuals with chronic hyperinsulinemia, this mechanism is impaired and insulin secretion pulsatility is progressively diminished. Since CEACAM1 is the main factor for insulin clearance, and impaired insulin clearance correlates more with hyperinsulinemia than with insulin secretion [86], it is safe to deduce that CEACAM1 impairment leads to a hyperinsulinemic state.

Interestingly, exogenous insulin delivery in the peripheral circulation is the only method of treatment for individuals with type 1 diabetes (T1D). Not only is it unlikely that all the exogenously administered insulin reaches the liver, but also this causes altered dynamic of insulin delivery, in opposition to endogenous insulin production [86,90]. This altered hepatic exposure to insulin is congruent with the loss of the insulin pulsatility effect, described above. Knowing the importance of the insulin pulsatility effect on the proper functioning of CEACAM1, CEACAM1 function could be impaired in patients with T1D [23,82,84].

However, some studies did not show an increased prevalence of NAFLD in patients with T1D compared to individuals without diabetes [4,91,92]. The age of the population studied, and the diagnostic methods probably played a role in these limited studies. However, the subject is still debated given the lack of studies in this field, which remains to be explored.

CEACAM1 also plays a critical role in hepatic lipogenesis. As explained, CEACAM1 in hepatocytes is activated through its phosphorylation mediated by the insulin–insulin receptor complex. Phosphorylated CEACAM1 binds to fatty acid synthase (FAS) and suppresses its activity by sequestering it. FAS is an enzymatic complex essential for enhancing fatty acids synthesis, and its inhibition by activated CEACAM1 leads to decreased liver lipogenesis [93].

Obese individuals have excessive white adipose tissue and, hence, high levels of circulating FFA [94]. FFA action on hepatocytes will promote and activate peroxisome proliferator-activated receptor α (PPARα) [95], which is a nuclear receptor acting as a ligand-activated transcriptional factor regulating the expression of crucial genes participating in fatty acid beta-oxidation [96]. PPARα reduces the activity of one of the CEACAM1 promoters, thus downregulating CEACAM1 protein transcription by decreasing CEACAM1-mRNA action [95]. Individuals with metabolic syndrome tend to have more FFA and thus have a much lower CEACAM1 activity. Whenever CEACAM1 activity is reduced by 50% or more, its sequestration of FAS is greatly diminished, and insulin clearance is also impaired inducing hyperinsulinemia, then leading to insulin resistance and to increased FAS activity [78]. This ultimately leads to steatosis and to NAFLD development, and puts forward the crucial importance of CEACAM1 protein in the metabolic balance and in the buildup of chronic fatty liver disease.

CEACAM1 protein synthesis in hepatocytes is promoted by the peroxisome proliferator-activated receptor γ (PPARγ) [97]. GLP-1, an incretin hormone, stimulates and upregulates PPARγ expression and activity [82,97], the activity of which has been shown to be reduced in patients with chronic hyperinsulinemia [98]. This suggests that reduced activity of GLP-1 implies a downregulation and, thus, a reduced activity of CEACAM1, but also implies that a treatment with GLP-1 receptor agonists could be a possible therapy to increase CEACAM1’s protective effect on the liver.

## 3. Discussion

In recent years, NAFLD has attracted increasing amounts of attention. It is known now that there are several complex biological mechanisms leading to NAFLD [2]. Understanding these mechanisms is key to help further research towards potential therapeutic targets.

Oxidative stress is a factor in the development of NAFLD and occurs when there is an imbalance between pro-oxidants and antioxidants. This can lead to cellular damage due to ROS. Treatments targeting oxidative stress, such as vitamins C and E in combination with statins, induce improved redox status in NAFLD patients. Coffee may help protect against NAFLD by regulating metabolism, fat accumulation, and providing polyphenol antioxidants [99]. Metformin appears to slow NAFLD progression by altering the gut microbiota and promoting ROS clearance through SOD. GLP-1 receptor agonists appear to have antioxidant effects by activating antioxidant genes. However, there are still multiple therapeutic approaches to be studied given the multi-mechanical roles of ROS in NAFLD pathophysiology [63].

CEACAM1, a transmembrane protein, is critical for many different types of processes, including tissue organization, metastasis, immune response control, and, more significantly here, metabolic homeostasis. It has been shown that its main effect regarding the liver is rather protective and its actions prevent hepatic fat deposition [82]. CEACAM1 is downregulated in individuals with the metabolic syndrome and its downregulation enhances a vicious circle leading to worsening of NAFLD. Since CEACAM1 plays a key role in the control of the metabolic aspects in NAFLD, its actions and the pathways it enhances could be interesting targets for therapeutic approaches. A stimulation of CEACAM1 through the known mechanisms (Figure 2) or through a new molecule is yet to be tested in NAFLD in order to assess its healing potential [63].

Despite all the current research performed in the field, there is no highly specific blood marker for NAFLD put in place yet. A blood marker or a constellation of blood markers related to CEACAM1 activity could potentially be interesting to help assess NAFLD presence and/or severity.

## 4. Conclusions

ROS and CEACAM1 activity are important factors to understand NAFLD development in individuals suffering from the metabolic syndrome (obesity, insulin resistance, T2D). Their understanding at the molecular level is crucial to unravel potential therapies targeting these different molecular pathways.

Restraining ROS activity via antioxidant therapy or via an antioxidant diet has shown promising results, giving hope for other molecular alternatives that reduce ROS activation. Regarding the antioxidant diet, it has been shown that coupled intake of vitamin E and vitamin C, two molecules having an antioxidant effect by eliminating free radicals, was beneficial on NAFLD and NASH. Coffee consumption also seems to protect against NAFLD although the mechanism is not fully understood, but studies appear to show a link with an antioxidant response. Other foods that are part of the antioxidant diet (polyphenols, carotenoids, glucosinolates) are being studied and appear to improve NAFLD [63]. There is significant evidence that CEACAM1 upregulation reduces fat deposition in the liver and optimizes insulin clearance, limiting the hyperinsulinemic state. To that matter, some potential molecules such as GLP-1 receptor agonists, used to treat T2D, or other molecules have yet to be further studied in order to judge their efficiency on NAFLD/NASH.

Understanding these molecular mechanisms involved in NAFLD development opens new therapeutic fields of interest, especially since NAFLD remains the most frequent chronic liver disease in the general population.

## Figures and Tables

**Figure 1 ijms-24-11271-f001:**
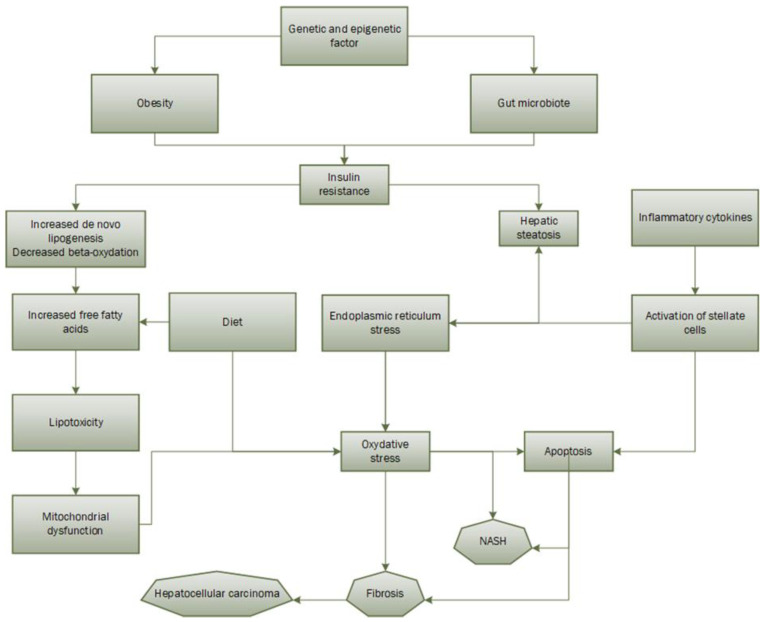
Multiple-hits hypothesis of NAFLD pathogenesis.

**Figure 2 ijms-24-11271-f002:**
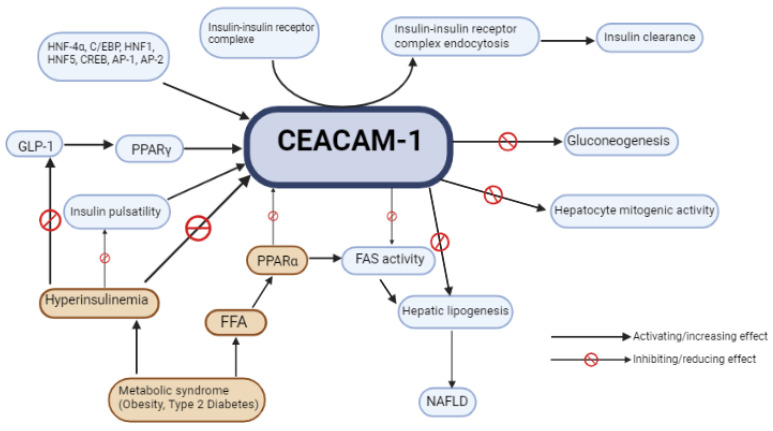
Roles and actions of CEACAM1 as well as factors influencing the functioning of CEACAM1.

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
