# Peer review of "Role of Oxidative Stress and Carcinoembryonic Antigen-Related Cell Adhesion Molecule 1 in Nonalcoholic Fatty Liver Disease"

_ijms, 2023, doi:10.3390/ijms241411271_

Round 1
Reviewer 1 Report
Memaj et al provided an informative review regarding the role of CEACAM1 and oxidative stress in NAFLD. However, there are still some points that can be improved.
Major:
· Line 23/24: NASH does not necessarily mean fibrosis. Please be more specific.
· Line 109-111: The causality here is probably the other way around
· Besides liraglutide, there are also clinical trials for another GLP-1 agonist (semaglutide) running. Please have a closer look at these and compare.
· I understand that ROS, especially from b-oxidation, does contribute to NAFLD, but how are the specific sources “radiation, certain drugs, or UV radiation” relevant for liver damage?
· Please correct the order of the figures. In the captions 2 comes before 1.
· Figure 2 (in reality 1): It is difficult to capture the complexity of NAFLD within one graphic but Fig. 2 lacks several important features, some of them even mentioned in the text. For example, you write in the text that damage of lipids by ROS results in damaged mitochondria, however, there is no arrow from ROS to FA damage or Mito stress, just the other way around. Where are the inflammatory cytokines coming from? In the diagram, NASH seems to be an end stage without connection to fibrosis. You might also emphasize “hepatic” insulin resistance/de novo lipogenesis. Stellate cells do not only have negative functions in NAFLD. Also, please indicate which of these steps are reversible.
In addition, you might consider placing diet at the top of the diagram, instead of genetic/epigenetic factors. Diet has a direct influence on obesity and gut microbiome. Genetic and epigenetic factors are also important but more by providing a susceptible background.
· Line 195: Probably you mean intracellular ROS levels not plasma levels
· Line 259: I am confused by the nomenclature. In paragraph 2.3.2 you distinguish between CEACAM1-3/4S/L and now you write CEACAM1-L. Please be more precise.
· Line 297-304: This paragraph gives the impression that T1D Patients are at very high risk to develop NAFLD, while in reality the disease prevalence in this group is even lower than in the general population. Please adjust.
· Line 302-304: “Knowing the importance of the insulin pulsatility effect on the proper functioning of CEACAM1, patients with T1D are probably more likely to have CEACAM1 impairment (82)”. This is a very strong statement that should not be based solely on your own review article.
· It would be interesting to see if CECAM1 functions better when T1D patients are treated with pulsatory insulin application. Are there any data on human or animal studies regarding this?
· Line 33-341: Citations are missing
· Line 367: “Limiting ROS activation” is not the best expression, as antioxidant treatments mostly do not inhibit ROS formation but rather capture and de-toxify ROS
· Figure 1 (in reality 2) is confusing. Why is CEACAM1’s inhibitory role on cell proliferation or Gluconeogenesis depicted as an activation of a reduction instead of simply a reduction? Why are there 2 shades of green?
The English language in general needs improvement.
Some specific points are:
The expression “pathophysiology” is more commonly used than “physiopathology”
Line 315: “PPARα reduces one of CEACAM1 promoter activity,” there seems to be missing a word
Author Response
Please see attached doc

Reviewer 2 Report
I have read a manuscript by Memaj et al., 2023 titled "Role of oxidative stress and carcinoembryonic antigen related 2 cell adhesion molecule 1 in nonalcoholic fatty liver disease"
This is a review that evaluated evidence from 94 studies that were comprehensively searched on different databases. The focus was more on oxidative stress and the key role of carcinoembryonic antigen-related cell adhesion molecule 1 (CEACAM1) in the in the physiopathology of NAFLD. They indicated that ROS and CEACAM1 activity are important factors in understanding NAFLD development especially in individuals suffering from the metabolic syndrome (obesity, insulin resistance, and T2D).
Some of the limitations observed are as follows:
(1) In the abstract, authors state their focus is on three main factors, including (1) oxidative stress, (2) insulin resistance, and (3) carcinoembryonic antigen-related cell adhesion molecule-1 this is quite different from what they mentioned in the introduction. These contradicting statements make it difficult for readers to follow their work.
(2) The introduction is too brief and not fully informative. As outlined by their aim "This study will focus on two significant 26 processes that contribute to the pathogenesis of NAFLD, namely oxidative stress and carcinoembryonic antigen-related cell adhesion molecule 1 (CEACAM1)". It would be important for the manuscript to focus on these two aspects. For instance, the information presented from 2.11 to 2.1.5 should be summarised and form part of the introduction which currently is not detailed. Although some information will still be relevant they can form part of the discussion.
(3) Line 137: use ROS as it is already written in full.
(4) Line 134: The subheading used are not informative perhaps you could remove; for example, this information is not about defining OS. but shows its level in both normal and in case of imbalance. perhaps you could rephrase the heading or remove them.
(5) line 141: perhaps all this information can be presented nicely in a table form.
(6) Figure 2 which presents the pathogenesis of NALFD needs revision, this NAFLD does not feature in this figure. I also recommend other platforms that can assist you in creating more appealing figures, for example, Biorender (https://www.biorender.com/). However, this is only a suggestion and not an advertisement of the platform.
(6) line 360: Correlated, rephrase this.
(7) line 367: Appropriate diet? What do you mean by this?
(8) In the concluding statement, statements made seem to be cited. I would suggest to authors to give a summary of findings from this review instead of citing other scholars in their concluding statement.
Author Response
Please see attached doc.

Round 2
Reviewer 1 Report
Thanks for improving the manuscript.
Please make sure to number all references according to their appearance in the text.
Reviewer 2 Report
A revised manuscript has improved however, some concerns still remains.
For instance figure quality, this figure is not clear for readers to interpret.
The authors states in their rebuttal letter that they used Biorender, and this is not the case based on the quality of generated figure 2.
From "appropriate diet " to "antioxidant diet" what does this mean?
The use of colour to distinguish between reducing and activation is not inclusive. Again when a reader print such in " black and white paper" this might not be informative. I suggest they use a different approach, maybe the size of arrows or anything else.
Round 3
Reviewer 2 Report
Thanks for revising the work, although some revisions were made, it still not clear what the authors have done regarding Figure 1 as I still cannot interpret it is not clear. Perhaps they can improve this quality as done with Figure 2.
Author Response
The quality of the figure was indeed suboptimal. We now improved it but think the Figure is clear to understand. If more resolution/work should be done on Figure 1, we will need help form the editorial assistants.
